# *Mycobacterium tuberculosis* Shikimate Pathway Enzymes as Targets for the Rational Design of Anti-Tuberculosis Drugs

**DOI:** 10.3390/molecules25061259

**Published:** 2020-03-11

**Authors:** José E. S. Nunes, Mario A. Duque, Talita F. de Freitas, Luiza Galina, Luis F. S. M. Timmers, Cristiano V. Bizarro, Pablo Machado, Luiz A. Basso, Rodrigo G. Ducati

**Affiliations:** 1Centro de Pesquisas em Biologia Molecular e Funcional (CPBMF) and Instituto Nacional de Ciência e Tecnologia em Tuberculose (INCT-TB), Pontifícia Universidade Católica do Rio Grande do Sul (PUCRS), Porto Alegre, RS 90619-900, Brazil; jeduardo.sacconi@gmail.com (J.E.S.N.); alejandro.duquevillegas@gmail.com (M.A.D.); talita.freitas@acad.pucrs.br (T.F.d.F.); luizagalina89@gmail.com (L.G.); cristiano.bizarro@pucrs.br (C.V.B.); pablo.machado@pucrs.br (P.M.); 2Programa de Pós-Graduação em Medicina e Ciências da Saúde, PUCRS, Porto Alegre, RS 90619-900, Brazil; 3Programa de Pós-Graduação em Biologia Celular e Molecular, PUCRS, Porto Alegre, RS 90619-900, Brazil; 4Programa de Pós-Graduação em Biotecnologia (PPGBiotec), Universidade do Vale do Taquari (Univates), Lajeado, RS 95914-014, Brazil; luis.timmers@univates.br; 5Centro de Ciências Médicas (CCM), Univates, Lajeado, RS 95914-014, Brazil

**Keywords:** *Mycobacterium tuberculosis*, human tuberculosis, shikimate pathway, enzyme drug target, enzyme inhibition, rational drug design

## Abstract

Roughly a third of the world’s population is estimated to have latent *Mycobacterium tuberculosis* infection, being at risk of developing active tuberculosis (TB) during their lifetime. Given the inefficacy of prophylactic measures and the increase of drug-resistant *M. tuberculosis* strains, there is a clear and urgent need for the development of new and more efficient chemotherapeutic agents, with selective toxicity, to be implemented on patient treatment. The component enzymes of the shikimate pathway, which is essential in mycobacteria and absent in humans, stand as attractive and potential targets for the development of new drugs to treat TB. This review gives an update on published work on the enzymes of the shikimate pathway and some insight on what can be potentially explored towards selective drug development.

## 1. Introduction

### 1.1. Tuberculosis

*Mycobacterium tuberculosis* is the main causative agent of human tuberculosis (TB). In 2018, approximately 10 million people developed TB, which resulted in 1.3 million deaths in HIV-negative and 300,000 deaths in HIV-positive patients. This recent report from the World Health Organization (WHO) indicated that, worldwide, approximately 400,000 people developed multidrug-resistant TB (MDR-TB), where *M. tuberculosis* strains are resistant to isoniazid and rifampicin, two of the most effective TB first-line drugs [1]. Even more concerning was the fact that 8.5% of MDR-TB patients were infected with extensively drug-resistant strains of *M. tuberculosis* (XDR-TB), MDR-TB strains that are resistant to any of the fluoroquinolones and to at least one of the three injectable second-line drugs. Hence, TB drug resistance represents a public health crisis and health security threat. The current treatment recommended by the WHO for cases of drug-susceptible TB is a 6-month regimen of four first-line drugs (isoniazid, rifampicin, ethambutol and pyrazinamide), with treatment success rates of at least 85%. MDR-TB requires a longer and more complex treatment regimen, with more expensive and toxic drugs, and has treatment success rates of 56% [1].

Worldwide, approximately 1.7 billion people are estimated to have latent TB infection (patient is infected with *M. tuberculosis* but has no symptoms), caused by the presence of “persisters”, being thus at risk of developing active TB during their lifetime [1]. Persisters are defined by a quiescent (non-growing or slow-growing) subpopulation of organisms that survive exposure to a bactericidal antibiotic, are genetically indistinct from drug-susceptible bacteria and can revive under antibiotic-free conditions [2]. They are associated with reduced metabolic rate, activated stress response and altered cell-wall permeability when compared to drug-susceptible bacilli, and are primarily established in macrophages or granulomatous lesions inside the human host. Besides being associated with latent infection, persisters are thought to contribute to the requirement for lengthy anti-TB treatment and to play a significant role in relapse [2]. Patients having latent TB represent a considerable reservoir of the bacilli.

The two main interventions employed to prevent new *M. tuberculosis* infections and their progression to active TB are the treatment of latent TB infection and the vaccination of children with the bacille Calmette-Guérin (BCG) vaccine [1,3]. Unfortunately, vaccination against sensitive and resistant strains of *M. tuberculosis* is not effective in preventing pulmonary TB in adults. Furthermore, the efficacy of BCG vaccination against pediatric pulmonary TB ranges from no protection to very high protection (0-80%) [4]. New strategies are thus needed to combat TB worldwide.

### 1.2. Drug Screening

The publication of the complete *M. tuberculosis* genome sequence [5] offered opportunities to implement screening campaigns focused on the identification of low molecular mass chemical compounds that inhibited the activity of target enzymes. These protein targets were selected based on gene essentiality and were purified, crystallized and had their binding sites identified. Although target-based enzymatic assays, which evaluate in vitro inhibition of protein activity and small chemical compounds with lower inhibition dissociation constant values (larger affinity values), are selected based on the assumption that they will be translated into target specificity, these efforts have not yielded new anti-TB agents, as target-based in vitro screening of enzyme inhibitors neglect essential factors, including cell wall permeability, metabolic stability and target vulnerability [6]. With few exceptions that confirm the norm, current antibacterial drugs and compounds in clinical trials target well-established functions in macromolecular synthesis, namely, cell wall, protein and nucleic acids [7]. Indeed, all current TB drugs were discovered in whole-cell screens for inhibition of *M. tuberculosis* growth or growth of a surrogate of *M. tuberculosis* [6].

Accordingly, there has been a revival on the interest of phenotypic drug discovery approaches based on their potential to address the incompletely understood complexity of diseases, their promise of delivering first-in-class drugs and major advances in the tools for cell-based phenotypic screening [8]. Ideally, phenotypic drug discovery campaigns utilize physiologically relevant models that are linked to patient-derived biology, which in turn may self-select small molecule phenotypic hits affecting pathways and protein targets most relevant to the disease of interest [9]. Although phenotypic drug screening interrogates a larger molecular target space, primary hits are rarely target-specific [8,10]. It has been pointed out that the classification of drugs as “phenotypically discovered” is somewhat inconsistent and that the majority of successful drug discovery programs combine target knowledge and functional cellular assays to identify drug candidates with the most advantageous molecular mechanism of action [8]. A balanced approach thus appears to be more appropriate for chemotherapeutic development of anti-TB agents. Incidentally, a thorough review on target-based whole-cell phenotypic screening strategies for anti-TB drug discovery has been reported [6]. Target-based whole-cell screening is designed to identify compounds that can penetrate the cell envelope and inhibit the target protein within the cellular environment.

Advantages of the target-based approach to the development of new chemical entities (NCEs) to treat TB are as follows: 1) the ability of performing in vitro high-throughput screening campaigns, 2) the search and identification of lead compounds with defined molecular mechanisms against a defined target, 3) the analysis of compounds with a favorable cost/benefit ratio, 4) the development, even in the initial stages, of compounds with selective toxicity (e.g., enzymes of essential pathways that are absent from the human host), 5) structural data of the target, when available, help guide medicinal chemistry efforts to improve pharmacodynamics and pharmacokinetics, 6) the pre-clinical evaluation of lead compounds and 7) generation of patents if a new chemical scaffold is proposed. Moreover, phenotypic screening efforts may be followed by identification of target(s) and, if amenable, heterologous expression, purification, protein assay and structural determination to help guide medicinal chemistry efforts (closing full circle). In both cases, small molecule prioritization from target-based drug discovery and phenotypic drug discovery would benefit from a deeper appreciation of the mode of action and target engagement profile [9].

### 1.3. Preferable Characteristics of Targets and Enzyme Inhibitors/Anti-TB Agents

Amongst limitations of the target-based approach is that the target may not be relevant in the disease pathogenesis setting, thus placing a major emphasis on target validation. An ideal target for antibiotic development should be essential for in vivo growth of the pathogen, be drug vulnerable and be druggable. A typical first step to establish essentiality of a gene product is genome manipulation of bacilli (knockout experiments). Interestingly, mycobacterial CRISPR interference has recently been employed for the rapid validation of target essentiality and compound mode of action [11]. However, not all essential gene products are equally vulnerable to drug mode of action and the target must be accessible to chemical inhibition [6]. An essential gene product that requires drug binding to 100% of intracellular protein target sites is not vulnerable as the required high target engagement to effect a physiological change is unlikely to be achieved in the intracellular context. Therefore, a druggable target should be both essential and vulnerable.

A druggable target can be reached and influenced by a particular chemical compound and the characteristics of its cognate-compound binding sites are as follows—be of reasonable size, enclosure (not extensively exposed to solvent) and hydrophobic with unexceptional hydrophilicity (sites with increased polarity are less likely to be druggable). The target site size is an important indicator of druggability, as the larger the better for opportunities to make chemical modifications to optimize physicochemical properties (e.g., solubility) without affecting ligand binding affinity to protein target [12]. It is desirable that a three-dimensional structure of a protein target be available and, in the case of enzymes, that a direct and continuous method is feasible for in vitro biological activity determination (assayability) to enable high-throughput screening of chemical compounds [13]. If the target is an enzyme, it should bind the chemical compound with larger affinity than substrate(s), the chemical compound should inhibit enzyme activity at concentrations that are compatible with its physicochemical properties (e.g., solubility) and lead to cell dysfunction (e.g., attenuated growth or cellular killing).

Enzyme inhibitors and inactivators comprised roughly half of all marketed drugs [14]. A thorough understanding of the chemical and catalytic mechanisms of a protein target is pivotal to guide efforts of medicinal chemists to improve target affinity. Understanding the course of an enzyme-catalyzed chemical reaction can help conceptualize different types of inhibitors and inform the design of screens to identify desired mechanisms [15]. The determination of the type of inhibition of chemical candidates is needed as there are preferable modes of action of enzyme inhibitors for hits to move forward in optimization efforts.

Ideally, new chemical agents to treat TB should [6,7]: 1) be effective against MDR- and XDR-TB (preferentially have a novel mechanism of action), 2) shorten treatment time (targeting nonreplicating bacilli and being effective in sterilizing the diverse subpopulations of *M. tuberculosis* in the human host), 3) be compatible with current antitubercular and HIV therapeutics (since TB/HIV co-infection is prevalent), 4) have favorable pharmacokinetic and pharmacodynamics profiles (manageable host metabolism profile and good oral bioavailability), 5) be chemically stable, 6) have appropriate physicochemical properties (e.g., aqueous solubility and absence of polymorphs), 7) have low cost of synthesis and as ultimate aspiration, 8) achieve a stable and relapse-free cure of TB. Interestingly, it has been proposed that targeting host lipid droplets in foamy macrophages may enhance efficacy of anti-TB agents. The lipophilic antibiotics stored in lipid droplets would be transferred to the intracellular pathogen as *M. tuberculosis* consumes host lipid droplets as a carbon source for growth [16].

The following section gives a description of the *M. tuberculosis* shikimate pathway enzymes as targets for the rational design of chemotherapeutic agents to treat TB. All the data was obtained from experimental results published in international journals, the great majority of which being indexed to PubMed. General keywords used for the searches were a combination of “enzyme name”/shikimate pathway, *Mycobacterium tuberculosis*, enzyme inhibition, (enzyme) drug target, etc. In this review, we have covered publications ranging from 1954 to 2019.

## 2. The Shikimate Pathway

The biosynthesis of aromatic rings from carbohydrate precursors in microorganisms and plants involves a range of extraordinary chemical transformations that together constitute the shikimate pathway; through seven enzymatic steps (Figure 1), phosphoenolpyruvate (PEP) and D-erythrose 4-phosphate (E4P) are condensed to the branch point compound chorismate (end product), which leads to several additional terminal pathways [17].

The shikimate pathway is an attractive target for the development of herbicides and antimicrobial agents because it is essential in algae, higher plants, fungi, bacteria and apicomplexan parasites but absent from mammals [18,19,20,21,22]. The mycobacterial shikimate pathway leads to the biosynthesis of chorismic acid, which is converted by distinct enzymes to prephenate (precursor of phenylalanine and tyrosine), anthranilate (precursor of tryptophan), aminodeoxychorismate (precursor of *para*-aminobenzoic acid—PABA—which, in turn, leads to tetrahydrofolate synthesis), *para*-hydroxybenzoic acid (precursor of ubiquinone or Coenzyme Q) and isochorismate (common precursor of naphthoquinones, menaquinones and mycobactins) (Figure 2).

The mycobacterial genetic determinants (the “counteractome”) allow these cells to overcome the host defense, which attempts to starve mycobacteria of tryptophan by a CD4 T cell-mediated killing mechanism. *M. tuberculosis*, however, can synthesize tryptophan (from chorismate precursor) under stress conditions and, thus, starvation fails [23]. Accordingly, inhibition of any enzyme of the mycobacterial shikimate pathway should preclude tryptophan biosynthesis and thereby increase the likelihood of starvation as an efficient mechanism of killing afforded by the human host.

### 2.1. 3-Deoxy-D-Arabino-Heptulosonate-7-Phosphate Synthase (aroG Coding Sequence; EC 2.5.1.54)

The first component enzyme in the shikimate pathway (Figure 1), 3-deoxy-d-arabino-heptulosonate-7-phosphate synthase (DAHPS), catalyzes the first committed step in the biosynthesis of aromatic compounds in plants and microorganisms. This enzyme is responsible for an aldol condensation reaction between PEP and E4P to produce 3-deoxy-d-arabino-heptulosonate 7-phosphate (DAHP) and inorganic phosphate [24,25]. DAHPS enzymes are currently classified into two distinct families—type I and type II—based on their amino acid sequence and molecular mass. Type I enzymes have a molecular mass of approximately 40 kDa and can be divided into two sequence subfamilies, Iα and Iβ. Type II enzymes are larger, with molecular masses over 50 kDa. Although there is less than 10% sequence identity between types I and II, all DAHPS enzymes share a common (βα) TIM barrel catalytic domain, with extra-barrel elements related to allosteric function [26,27,28]. DAHPS activity is allosterically regulated by all three aromatic amino acids produced by the pathway—phenylalanine (Phe), tyrosine (Tyr) and tryptophan (Trp)—therefore being the major flow control point for the shikimate pathway [17,26].

In *Escherichia coli* there are three structural genes—*aroF*, *aroG* and *aroH*—coding distinct type Iα DAHPS enzymes, each has different extensions on (β/α) barrel domain and an N-terminal barrel extension, which provide a single allosteric binding site that is selective for either Phe, Tyr or Trp [17,29,30]. *Thermotoga maritima* possesses a single DAHPS type Iβ, which contains an N-terminal domain associated with a single binding site for Tyr. In *M. tuberculosis*, sequence homology analysis revealed a single gene predicted as *aroG* (1389 bp), coding a type II DAHPS enzyme (*Mt*DAHPS) [31]. Overall, the catalytic activity of DAHPS depends on the presence of a divalent metal ion in the active site. The activity of EDTA-treated apo *Mt*DAHPS could be restored in assays containing Co^2+^, Mn^2+^, Cd^2+^, Cu^2+^, Zn^2+^ and Ca^2+^, with Co^2+^ and Mn^2+^ yielding the best results for restoring activity [28]. Furdui and coworkers [32] conducted experiments demonstrating that, in the case of *E. coli* DAHPS, the metal ion in the active site of the enzyme seems to play a structural role, orchestrating the arrangement of the active site residues in a position favorable for water activation. In this view, the geometry coordination of different metal ions may be a valuable factor for rational inhibitor design.

Previous studies have shown that the *Mt*DAHPS reaction follows an ordered sequential mechanism, similar to those described for *P. aeruginosa*, *Helicobacter pylori* and *Xanthomonas campestris* type II DAHPS enzymes characterized to date [28,33,34]. The chemical mechanism described begins with a nucleophilic attack of PEP at the E4P aldehyde moiety, resulting in the formation of an oxocarbenium species **A** (Figure 3), which can be attacked by an active site water to form a phosphohemiketal intermediate **B**. This water molecule can potentially attack from either face of the oxocarbenium ion **A**, giving rise to two possible diastereoisomers of the tetrahedral intermediate phosphohemiketal **B**, differing in their absolute configuration at C2. Although this stereogenic center is transient and the stereochemical formation is lost by the elimination of phosphate in the final step to generate the product DAHP **C**, the geometry of the enzyme’s active site is likely to favor stereoselective attack of water to form one diastereoisomer tetrahedral intermediate phosphohemiketal **A**, preferentially. Thereby, DAHP is formed in its acyclic form and cyclizes into its cyclic pyranose form following release from the enzyme’s active site [35].

The crystallographic structure of *Mt*DAHPS (PDB 2B7O) revealed a homotetrameric quaternary structure whose dimeric and tetrameric interfaces are formed by accessory interactions between (β/α) 8 TIM barrel subunits (Figure 4a) [28]. The structure also enabled the identification of possible catalytic (and essential) amino acid residues at the active site revealing a network of hydrogen bonds between phosphate and carboxylate groups from the protein and PEP, which defines the binding site for the substrate (PEP). The analysis of the structure showed that the PEP phosphate group makes hydrogen bonds with the NH_2_ group of Lys306, the δ-guanido groups of Arg284 and Arg337, the amide nitrogen of Glu283 and with two water molecules (Figure 4b). The PEP carboxylate group forms a double hydrogen bond with the δ-guanido group of Arg126 and one additional hydrogen bond with the ε-amino group of Lys306 and one water molecule. Thus, each carboxylate oxygen atom receives two hydrogen bonds with favorable geometry for the molecular recognition of substrates at the enzyme’s active site (Figure 4b). Furthermore, a sulfate ion (SO_4_^2-^) is bound in the active site region near its opening and approximately 10 Å away from PEP. This ion marks the likely position of the phosphate group from E4P and makes hydrogen bonds with amino acid residues Arg135, Arg284 and Ser136. Arg284 provides a bridge between PEP and sulfate, where there is one hydrogen bond between the residue and the PEP phosphate group and another hydrogen bond between the residue and the sulfate ion. The metal binding site is formed by four ligands—the Sγ of Cys87 (from the β1-α1 loop), Nξ2 of His369 (from the β7-α7 loop), Oξ2 of Glu411 (from the end of strand β8) and Oδ2 of Asp441 (from the β8-α8 loop). Cys87 and His369 are the axial ligands, with Glu411 and Asp441 in the equatorial trigonal plane, leaving one equatorial site available for the binding of E4P or water. An interesting feature of the metal site is the presence of Cys440 close to the metal ligand Cys87. A simple rotation at Cys440 Cα-Cβ to another rotamer would allow formation of a disulfide bond with Cys87 precluding metal binding (Figure 4b). This explains the need for a reducing agent to maintain full activity of the enzyme [28]. However, the importance of the amino acid residues mentioned above for the catalytic mechanism of *Mt*DAHPS has not been experimentally demonstrated and the knowledge of their identity can be important not only for understanding enzyme function but also for potential use in drug design.

Previous studies have shown that *Mt*DAHPS possesses a highly sophisticated mechanism of allosteric control due to an N-terminal extension and a loop extension. These extra-barrel elements provide three distinct binding sites that are selective for either Phe, Tyr or Trp. Site 1, occupied by Trp, is located on the tetrameric interface and consists mainly of the amino acid residues of the α2β3 loop extension and the helix α1. Site 2 is occupied by Phe and/or Tyr and is on the dimeric interface, covered by sheet β0 of the TIM barrel N-terminal extension. Site 3 is located between the N-terminal extra-barrel portion (α0a and α0b helices) and the α3 helix of the main barrel; Phe and/or Tyr occupy this site at high concentrations (2 mM). Even though not inhibited by individual aromatic amino acids, binary combinations with Trp and either Phe or Tyr result in significant loss of *Mt*DAHPS activity and the ternary combination of all three aromatic amino acids completely abolishes the enzyme-catalyzed chemical reaction, suggesting synergy in response to these allosteric effectors [36,37]. Furthermore, *Mt*DAHPS also forms a non-covalent hetero-octameric complex with *M*. *tuberculosis* chorismate mutase (*Mt*CM), which acts at the branch point that connects the shikimate pathway to Phe and Tyr production [26,37,38]. This complex formation not only activates *Mt*CM activity by more than two orders of magnitude but also allows *Mt*CM to share the allosteric machinery located on *Mt*DAHPS to direct the shikimate pathway end product—chorismate—towards either Trp or Phe and Tyr biosynthesis, based on metabolic requirements [39,40]. Interestingly, this remote control between DAHPS (the first enzyme of the shikimate pathway) and chorismate mutase represents a novel paradigm for regulation of this pathway [38].

Even though few inhibitor studies have been conducted targeting the mycobacterial enzyme, based on the crystal structure of *Mt*DAHPS (PDB 2B7O), Reichau and coworkers [35] designed a bisphosphate molecule as a simplified analogue of the tetrahedral intermediate (Figure 5a) as a novel lead structure for inhibitors of *Mt*DAHPS. This bisphosphate compound binds with its 2-phosphoryl and carboxylate moieties in the PEP binding site and in an extended conformation in the phosphate moiety on C7, located near the proposed E4P phosphate binding site. The bisphosphate forms salt bridges with Arg284, Arg337 and Lys306 and a hydrogen bond to the backbone N-H of Glu283. The carboxyl group forms salt bridges with Lys133 and Arg126 and coordinates to the manganese ion present in the active site. The 7-phosphate moiety extends into the tentative E4P phosphate binding site, forming salt bridges with Lys380 and Arg135 and hydrogen bonds with the Ser136 hydroxyl group and backbone N-H. Arg135 and Ser136 interact with the 7-phosphate moiety of the inhibitor, moving closer into the active site pocket. The side chain of Lys133 and the likely electrostatic repulsion with the guanidinium group of Arg135 shows the biggest conformational change of all active site residues, establishing a salt bridge to the carboxylate group of the bisphosphate. Synthesized enantiomer compounds (R)-bisphosphate and (S)-bisphosphate were used on inhibition assays, where inhibitory constants of 360 ± 50 nM and 620 ± 110 nM, respectively, were determined. These results provide valuable insight to a lead structure for novel potent antimycobacterial drugs targeting *Mt*DAHPS [35].

Furthermore, structure-based approach and E-pharmacophore based virtual screening identified 3-pyridine carboxyaldehyde, α-tocopherol and rutin as lead compounds to inhibit *Mt*DAHPS (Figure 5). Enzyme inhibition assays against *Mt*DAHPS confirmed their inhibitory potential, suggesting that their structure could be exploited for future drug development [41]. Desmond stimulation interaction diagram for α-tocopherol showed its interaction with Ser136, Asp138 and Gln37. For 3-pyridine carboxyaldehyde, there was a stable interaction with Arg399 and Lys 306. Rutin showed constant H-bond interaction with Arg284 andArg337. The percentage of enzyme inhibition for the lead compounds, α-tocopherol, 3-pyridine carboxyaldehyde and rutin, were 21%, 17% and 16%, respectively [41]. However, further efforts are needed to show whether these chemical compounds reach and inhibit *Mt*DAHPS in the intracellular context and target engagement results in arrested growth of bacilli.

### 2.2. 3-Dehydroquinate Synthase (aroB Coding Sequence; EC 4.2.3.4)

The second enzyme in the shikimate pathway (Figure 1), 3-dehydroquinate synthase (DHQS), is responsible for the generation of the first carbocyclic compound of the pathway, converting DAHP into 3-dehydroquinate (DHQ) [21,42,43]. In *M. tuberculosis*, DHQS is encoded by the *aroB* gene, differing from microbial eukaryotes where this homologue is part of a multifunctional enzyme complex (encoded by the AROM gene) capable of conducting the five central steps of the pathway [44,45,46]. The mycobacterial enzyme has the extraordinary ability to conduct six chemical reactions (Figure 6) in a single active site, which was considered by some a marvel feat, attracting interest on the enzyme’s structure and chemical mechanism [46,47,48].

The cloning and functional demonstration of *Mt*DHQS activity was first reported by Mendonça and coworkers [49] and studies on its kinetic mechanism were subsequently performed [50]. Initial velocity measurements and fluorescence binding studies point to a rapid equilibrium ordered mechanism, with *K*_M_ values of 6.3 μM for DAHP and 70 μM for NAD^+^. DHQS is a metalloenzyme, requiring a divalent cation for its activity. Incubation with EDTA abolishes enzyme activity, which can be restored by addition of Co^2+^ into the reaction mixture and by Zn^2+^, Ca^2+^, Cd^2+^, Mg^2+^, Mn^2+^, Ni^2+^ and Ba^2+^ to a smaller extent [50]. Although Cobalt seems to be more catalytically relevant, inductively plasma atomic absorption analysis of recombinant *Mt*DHQS revealed the presence of Zinc in the enzyme preparation, probably due to the greater bioavailability of Zinc and in agreement with reports related to the DHQS domain of AROM from *Aspergillus nidulans* [51,52]. The kinetic constants for the cyclic reduction and oxidation of NAD^+^ and NADH, respectively, during the *Mt*DHQS catalyzed reaction, were determined using a stopped-flow instrument, under single-turnover experimental conditions. The pre-steady-state kinetic data demonstrated that there is a transient increase in absorbance associated with NADH formation followed by its oxidation back to NAD^+^ that occurs only in the presence of *Mt*DAHP-catalyzed conversion of DAHP to DHQ. In addition, the rates of NAD^+^ → NADH → NAD^+^ conversion are not rate limiting, since they are larger than the 0.63 s^−1^
*k*_cat_ value [50].

The chemical mechanism of *E. coli* DHQS was thoroughly investigated, dissecting each of the steps using substrate analogues to examine the early and late steps of the reaction, which are—binding to the enzyme, C5 oxidation by enzyme-bound NAD^+^, C6 enolization, β-elimination of the phosphate group, reduction at C5 regenerating NAD^+^, pyranose ring opening, and, finally, an aldol intramolecular reaction [47,48]. A set of analogue compounds that were uncapable of undergoing β-elimination revealed that the enzyme binds with higher affinity to shortened side chains and carbocyclic analogues. For example, a single carbon reduction in the side chain rendered a three order of magnitude improvement on inhibition, which agrees with a previous report [53]. Carbocyclic compounds seem to have an increased affinity for the enzyme due to the favored oxidation at C5, as shown by the levels of NADH found in the enzyme-analogue complexes when in equilibrium. The C6 enolization is proposed to be carried out by the attack of one of the phosphate oxygens present in the substrate (DAHP), acting as a base and extracting a proton from C6, preparing the group for the subsequent elimination step and giving evidence for a stepwise reaction [47].

Another set of analogues, uncapable of interconverting between the hemiketal and acyclic ketal proposed intermediates, was used to demonstrate that the enzyme is able to use a 2-deoxy analogue of DAHP as substrate, which generates a product that does not go through the last two reactions (pyranose ring opening and aldol intramolecular reaction). Additionally, the enzyme does not seem to be able to bind to acyclic analogues of DAHP. This data implicates that the enzyme may not be actively involved in these last two steps and that, instead, they occur spontaneously (either in solution or in the active site) [48]. Altogether, these data suggest that the enzyme is probably specialized in the redox reaction at C5, which triggers a cascade of spontaneous events that generate DHQ, opposing an alternative hypothesis that this enzyme would have evolved to gather six catalytic groups into a single active site to carry out all the steps of the reaction. In fact, the structure of the AROM DHQS domain from *A. nidulans* shows that the enzyme coordinates a proton shuffling from the C5 hydroxyl group to a water molecule and then to His275 in a concerted manner while the hydride transfer to NAD^+^ occurs. A phosphate binding pocket positions the phosphate group so that one of the phosphate oxygens from DAHP can extract the proton from C6, resulting in the elimination of phosphate. There is little evidence for enzyme participation in the last two steps, corroborating the data from the study with substrate analogues [46].

Most of the analogues synthesized in the aforementioned studies [47,48] turned out to be competitive inhibitors of *Mt*DHQS, although there is no report on probing their potential as drugs to target bacteria. Nonetheless, a recent publication revealed that DHQS is the main target of IMB-T130 (Figure 7) [54]. This promising molecule exhibited an IC_50_ of 2.7 μM and a MIC of 312 ng/mL against *M. smegmatis*, a 1.36 log reduction in the growth of intracellular mycobacteria infecting mouse macrophages, with low toxicity towards human embryonic kidney 293 and HeLa cells [55], showing this enzymes’ potential as a druggable candidate.

### 2.3. 3-Dehydroquinate Dehydratase (aroD Coding Sequence; EC 4.2.1.10)

Dehydroquinate dehydratase—also known as dehydroquinase (DHQase)—was originally identified in *E. coli* and is the third enzyme in the shikimate pathway (Figure 1), catalyzing the dehydration of DHQ into 3-dehydroshikimate (DHS) [56]. This enzyme possesses two distinct classes (types I and II) often correlated with the role in two different pathways—type I is present in the chorismate biosynthetic pathway from fungi, plants and some bacteria, while type II was considered part of the catabolic quinate pathway of fungi and in the shikimate pathway of most bacteria [57]. *Mt*DHQase is one of the enzymes that indicated this classification could be misleading, as it is in fact a type II DHQase and part of the biosynthetic shikimate pathway [58]. Interestingly, types I and II DHQases seem to have evolved independently and are not related in sequence, structure or chemical mechanism [57,58,59,60,61].

Each type has its own stereospecificity [59,60,62,63]. Type I DHQases catalyze the dehydration with loss of the less acidic pro-*R* hydrogen bound to C2, with a covalently bound intermediate, a Schiff base formed between the C3 carbonyl group from the substrate and a conserved active site lysine followed by an enamine formation. A general base in the enzyme then extracts the proton from C2 and accepts the hydroxyl group in C1 leading to a *syn*-elimination [60,64,65]. In contrast, type II removes the more acidic pro-*S* hydrogen in accordance with the favored *anti*-elimination observed in acid or base catalyzed conversion of DHQ into DHS [60,62,64]. There is no covalent intermediate formed in this mechanism. A first proposition of an enolate intermediate was suggested by isotope effects studies [66] but new evidence of a structural water molecule would favor the formation of an enol intermediate before the exit of the C1 hydroxyl group [61].

Structurally, type I enzymes are homodimers in solution, with an eight stranded α/β-barrel topology, a fold observed in other enzymes that form an imine intermediate [57,65,67,68]. *Mt*DHQase displays a five-stranded parallel β-sheet core flanked by four α-helices with a flavodoxin-like overall topology, being an homododecamer in solution (a tetramer of trimers) with 23 symmetry [57,69], in accordance with other type II DHQases from *Streptomyces coelicolor*, *H. pylori*, *Acinetobacter baumannii* and *P. aeruginosa* [61,70,71,72]. Tyr24 (*M. tuberculosis* numbering) is conserved amongst type II enzymes and is proposed to be the base responsible for the extraction of the axial pro-*S* hydrogen [61,73]. Site-directed mutagenesis experiments showed that Arg23 (*S. coelicolor* numbering, equivalent to Arg19 in *Mt*DHQase) is an essential amino acid for catalysis [74], although its role is not totally clear as it is present in a flexible loop composed by residues 19-24 (*Mt*DHQase numbering) and it seems to approximate Tyr24 after substrate-binding, lowering its pK_a_ and starting the reaction [75]. It is worth noting that *Mt*DHQase classification as type II is based on structural and theoretical predictions. Enzyme kinetic studies (steady-state, pre-steady-state, pH-rate profiles, isotope effects, surface plasmon resonance, spectrofluorimetry, microcalorimetry, mass spectrometry, borohydride trapping) on *M. tuberculosis aroD*-encoded (Rv2537c) DHQase should be pursued to offer a solid support for a chemical mechanism proposal and, consequently, for an adequate classification type to which this enzyme belongs.

An accurate distinction between types I and II is of critical importance in the design of inhibitors as the differences in active site and mechanism can be explored to develop highly specific inhibitors. Several reports on *Mt*DHQase inhibition presented molecules displaying *K*_i_ vales in the nanomolar range [73,76,77,78,79,80], all of them being competitive analogues of DHQ substituted in either C2, C3 or both. The interactions of these compounds with the active site of *Mt*DHQase are driven mostly by two main features: (1) the presence of groups capable of forming π-staking interactions with Tyr24, like benzyl and benzothiophene groups attached to C2 or C3 and (2) groups that displace Arg19 outside the active site, blocking its interaction with Tyr24 [75,80].

While the number of promising compounds with low *K*_i_ values is high and a solid support from structural data is available, the number of articles reporting effective inhibition on growth of *M. tuberculosis* is reduced. In a series of compounds generated by virtual screening with no resemblance to DHQ, two lead molecules were identified with IC_50_ values of 17.1 and 31.5 μM and MIC values against *M. tuberculosis* H37Rv of 25.0 and 6.25 μg/mL, respectively [69]. In another effort, a pharmacophore model was developed and screened against a library of natural compounds. Two molecules selected from the library that fulfilled all the characteristics of the model displayed some inhibition on *M. tuberculosis* growth with MIC values of 100 μg/mL [81]. Finally, another series of compounds based on DHQ monosubstituted on C3 or disubstituted on C2 and C3 generated nanomolar *K*_i_ values but had to be transformed into esters to display low μg/mL MIC values [79]. The authors propose that increased hydrophobicity helps molecules cross the mycobacterial cell wall. Examples of these compounds are given in Figure 8.

### 2.4. Shikimate 5-Dehydrogenase (aroE Coding Sequence; EC 1.1.1.25)

The NADPH-dependent reduction of DHS is carried out by shikimate dehydrogenase (SD), the fourth enzyme in the shikimate pathway (Figure 1), generating the central metabolite shikimate (SHK). In some organisms, this point of the pathway intersects with the quinate catabolic pathway, where a second enzyme of the SD family is found, quinate dehydrogenase (QD), which catalyzes the reduction of both quinate and SHK, with first reports of the activity of the two enzymes from *E. coli* (SD) and *Aerobacter aerogenes* (QD) dating from about the same year (1955 and 1954, respectively) [56,82]. The presence of QD gives microorganisms the ability to use quinate as a carbon source, generating protocatechuate and proceeding to the β-ketoadipate pathway [83].

In most bacteria, SD is encoded by the *aroE* gene, like in *E. coli*, *M. tuberculosis* and *Methanococus jannaschii* [83,84,85,86], while QD is encoded by *yidB* in *E. coli* [83], *qsuD* in *Corynebacterium glutamicum* [87], *QA-3* in *Neurospora crassa* [88] and *qutB* in *A. nidulans* [89]. In some fungi, such as *A. nidulans*, *N. crassa* and *Saccharomyces cerevisiae* and in the apicomplexan parasite *Toxoplasma gondii*, SD is part of the multifunctional AROM enzyme complex, while in some plants like *Arabidopsis thaliana* and *Populus trichocarpa*, the enzyme is found in a bifunctional enzyme complex, with the third step of the pathway DHQase [90]. Three other members of the SD family were identified as aminoshikimate dehydrogenase [91,92], SD-like enzymes [87,93] and AroE-like enzymes [94], though limited data is available on their mechanisms and substrate specificity. It is noteworthy that one pursuing the design of inhibitors for SD should be aware of the variety of members in this family and the possible mechanisms of resistance that may have already evolved in bacteria and plants as a result of either redundant activity or substrate preference, the latter of which cannot be translated into substrate specificity.

While sequence identity varies between family members, the structure is well conserved amongst them [90]. The structure of SD is comprised of two α/β architecture domains connected by a pair of α-helices. The substrate biding site is located at the N-terminus of the protein, shaped by a β-sheet (usually composed by six mainly parallel β-strands) surrounded by α-helices. The dinucleotide biding site located at the C-terminus displays a classic Rossmann fold. A comparison between the *aroE* and *ydiB* gene products from *E. coli* revealed small differences in the amino acid composition of the dinucleotide binding pocket that could explain variability of cofactor specificity found in the SD and QD enzymes and amongst other members of the family. NAD^+^ binding in YdiB (PDB 1O9B) is favored by two amino acid substitutions compared to AroE (PDB 1NYT), which create in YdiB a more neutral environment compared to the basic binding pocket found in AroE [83]. In the *Mt*SD models generated by our group, Arg149, Asn150 and Lys153 are present in equivalent positions to *E. coli* AroE Arg150, Thr151 and Arg154, which would explain the mycobacterial’s enzyme preference for NADPH [95,96]. In *Thermus thermophilus* crystal structures, the DHS/SHK binding pocket (in the deep groove created between the two domains) reveals a set of polar amino acids capable of making hydrogen bonds with the hydroxyl groups of C3, C4 and C5 and, likewise, the carboxyl group at C1 [97]. The contacts with the C1 carboxyl and C4 hydroxyl groups are considered to be the most important for substrate recognition and catalysis, as mutations in the *A. thaliana* bifunctional enzyme DHQase-SD and *S. epidermidis* AroE enzymes showed a marked increase in *K*_M_ and decrease in *k*_cat_ [98,99]. These residues where found to be conserved in the *Mt*SD structure model [95,96].

A few reports on the kinetic mechanism of SD are available. Three independent studies revealed a consistent finding that SD enzymes follow a sequential bi-bi mechanism—ordered for *Pisum sativum* and *M. tuberculosis* [95,100] and random for *S. aureus* [101]. However, kinetic data from the *P. sativum* enzyme suggests the binding of NADPH first, which is in contrast with the data from *Mt*SD where DHS is proposed to bind first. The reported *K*_M_ values for SHK and cofactor are typically in the range of 50 and 200 μM for bacterial SD enzymes (*Mt*SD SHK *K*_M_ = 50.2 μM) [85]. Turnover rates can vary greatly depending on the species, ranging from 361 s^−1^ in AroE from *Archaeoglobus fulgidus* to 1.5 s^−1^ in AROM from *A. nidulans* (*Mt*SD *k*_cat_ = 8.2 s^−1^) [85,102,103]. However, a better consensus is found considering the chemical mechanism. The hydride transfer from NADPH C4 to the DHS hydroxyl group on C3 is mostly conducted by two well conserved Lys and Asp residues, while a conserved Tyr seems to be important to position the cofactor in a catalytic competent manner [90,95,96,98,104,105]. Site-directed mutagenesis data suggest an important role in catalysis for both residues, as an impact on *k*_cat_ was observed in *A. thaliana* [98] and on *M. tuberculosis* [105]. Data from pH-rate profiles and isotope effects on *Mt*SD also indicates that the deprotonation of a residue with pK_a_ ~8.9 abolishes the activity of the enzyme [95]. Similar results were observed for *Aquifex aeolicus* [104]. Both research groups refer the conserved Lys or Asp as possible residues to explain the results. Solvent isotope effects also corroborate the transfer of a proton from solvent as being important for catalysis, indicating a probable acid-base catalysis mechanism, with a stereospecificity of the pro-*S* hydride from NAPDH C4 being transferred in a concerted manner [95], in contrast with SD partially purified from *E. coli*, which showed a preference for the pro-*R* hydrogen [106]. Figure 9 presents the (a) kinetic and (b) chemical mechanisms of *Mt*SD.

Multiple SD inhibitors have been reported with IC_50_ values ranging from ~3 to ~900 μM [101,107,108,109,110], with only two reports showing MIC values of ~500 μM for a polyphenolic compound against *P. putida* [108] or shikimic acid derivatives against *E. coli* [109] and virtual screening derived compounds against *S. aureus* SD with *K*_i_ values ranging from ~8 to ~100 μM [101,110]. There are three other reports on triazolothiadiazole inhibition of *Mt*SD presented IC_50_ values ranging from ~7 μg/mL to ~5 mg/mL and MIC values ranging from 0.25 to 8 μg/mL [111,112,113]. One of the drugs (named IMB-SD62) showed in vitro activity against MDR-TB and in vivo anti-TB activity with a 1.7 log reduction in the CFUs in lungs and a preliminary pharmacokinetic study was performed demonstrating some potential for this class of molecules [113]. They display different inhibition mechanisms, which is valuable as they can turn into scaffolds for the design of several inhibition strategies against SD enzymes. A recent effort to identify potential molecules that bind to *Mt*SD through virtual screening found three molecules (out of a library of 13,000 compounds) with favorable binding energies compared to SHK [114]. However, no experimental evidence of inhibition was presented to date. Examples of the different inhibitors are provided in Figure 10.

### 2.5. Shikimate Kinase (aroK Coding Sequence; EC 2.7.1.71)

Shikimate kinase (SK), the fifth enzyme in the shikimate pathway (Figure 1), catalyzes the conversion of SHK into shikimate-3-phosphate (S3P) using adenosine 5′-triphosphate (ATP) as the co-substrate. The binding of ATP and SHK in the active site is guided by a conformational change of the flexible loop, promoting an open-to-closed conformation, followed by a rearrangement of domain movement for SK enzymes [115,116]. The release of S3P from the active site depends on the interaction with three conserved arginine residues—Arg117, Arg136 and Arg58. Arg117 triggers the process by forming a strong electrostatic interaction with the C3 phosphate group of S3P. The product undergoes a destabilization in the active center and a conformational change occurs, which in turn leads to a large opening of the extended SHK binding (ESB) and LID domains [117]. The catalytic turnover between SHK and ATP was studied in the crystalline environment. Based on a series of crystallographic studies with the enzyme in the apo form, binary and ternary complex, this study suggested that the random sequential binding of SHK and nucleotides is correlated with domain movements [115].

In *E. coli*, SK exhibits two isoforms that are encoded by different genes—*aro**K* encodes for a shikimate kinase type I (SK I) and *aro**L* encodes for a shikimate kinase type II (SK II). These enzymes have low sequence identity when compared to the mycobacterial enzyme, presenting 30% and 46% for SK I and II, respectively [118]. Other microbial genomes indicate a single SK coding sequence. In *M. tuberculosis*, SK is encoded by the *aro**K* gene, producing an 18.6 kDa protein with 176 amino acids. Two significant differences were observed by Gu and coworkers [119] in the substrate-binding (SB) domain between SK I and SK II. The first is the replacement of a phenylalanine residue (Phe49 in *M. tuberculosis* SK; *Mt*SK) in SK I by a valine residue in SK II. The second difference is a shift of a glutamate residue (Glu54 in *Mt*SK) in SK I by a tryptophan in SK II. It is important to observe that Glu54 (in *Mt*SK) forms a hydrogen bond and a salt bridge with arginine (Arg58) and can help position the Arg58 guanidine group for SHK binding.

This enzyme belongs to the family of nucleoside monophosphate (NMP) kinases, a subfamily of the P-loop including the nucleoside triphosphate hydrolase superfamily [120,121]. This family undergoes large conformational changes during catalysis. A comparative study between *Erwinia chrysanthemi* SK (apo) and *Mt*SK bound to ADP (binary complex) structures suggested that significant conformational changes occur upon nucleotide binding [119]. It was proposed by Dhaliwal and coworkers [122] that the binding of ADP induces a large hinged movement of the LID domain over the active site with Arg110 and Arg117, which is essential for the interaction with the nucleotide and repositioning of the SB domain observed.

*Mt*SK presents an α/β architecture, showing five central parallel β-strands flanked by α-helices. The overall structure is divided into four domains [115]: (i) the ESB domain (residues 32-93), which enables rigid body movements, including the SHK binding (SB) subdomain (residues 32-61) that corresponds to the binding of NMP in NMP kinases, (ii) the nucleotide binding (NB) domain, responsible for the flexible regions constituting the nucleotide binding, formed by the phosphate binding P-loop (Walker-A motif; residues 9-17), adenine-binding (AB)-loop (residues 148-155) and the segment corresponding to residues 11-110, including α6 (residues 104-110), (iii) the LID domain (residues Gly112 to Asp124), responsible for opening and closing the catalytic site, allowing the interaction of the enzyme with substrates and products and (iv) the reduced core (RC) domain, covering NB domain and ESB segment (residues 62-93). Amongst all, the RC domain is the most rigid portion of the enzyme, whereas the LID and NB domains are the most flexible regions (Figure 11) [119,120,123].

Based on crystallographic studies, Hartmann and coworkers [115] described two distinct conformational states for *Mt*SK—open and closed. In the open conformation (open LID), SHK or ATP can bind to the enzyme independently, thereby inducing LID closure. SHK has the same orientation in all complexes with *Mt*SK. The SHK binding site is characterized by a surface, usually hydrophobic, with some residues that align into the cavity. Pro11, Phe49, Phe57, Gly79, Gly80, Gly81, Pro118 and Leu119 form a lipophilic part of the pocket. The carboxylate group of SHK is anchored by a salt bridge with the guanidinium group of Arg136 and the side chain of Asp34 forms a bidentate hydrogen bond with the O12 hydroxyl group of SHK.

Kinetic studies of fluorescence quenching on *E. chrysanthemi* SKII demonstrated that the binding of the first substrate (nucleotide) enhanced the affinity for SHK as second substrate [124]. Based on steady-state kinetics, fluorescence spectroscopy and isothermal titration calorimetry data for monomeric *Mt*SK, it was suggested that this enzyme follows a rapid-equilibrium random order of substrate binding (SHK or ATP) and ordered product release, in which S3P is released first, followed by ADP dissociation (Figure 12) [125]. Hartmann and coworkers [115] identified conformational changes in *Mt*SK complexes that suggest synergic effects in the random sequential binding of SHK and ADP.

Experimental and computational studies have shown that catalysis may occur under specific conditions. The LID and SB domains are tightly closed presenting two possible arrangement: (a) the guanidinium group of the essential arginine, located in the LID domain, is close of the γ-phosphate of ATP for its activation and (b) the two substrates are very close together but isolated from the solvent environment for the reaction. The release of products requires a large conformational change involving the LID and SB domains, where the hydrogen bonds amongst the essential arginines (Arg117, Arg136 and Arg58) and the products need to be disrupted to release S3P [117,126].

*Mt*SK is an interesting target for the development of inhibitors, since its function is essential for the survival of *M. tuberculosis* [31]. Different researches have been making efforts to find a potent molecule against this enzyme, which could help in the treatment of TB. Recently, six new compounds were characterized as *Mt*SK inhibitors—manzamine A, 8-hydroxymanzamine A, manzamine E, manzamine F, 6-deoxymanzamine X and 6-cyclohexamidomanzamine A. Inhibition studies demonstrated that these molecules show a mixed noncompetitive type of inhibition [127]. Another study showed a collection of 14 hits (out of 404 compounds) containing oxadiazole-amide and aminobenzothiazole scaffolds, which displayed >90% inhibition at concentrations below 50 µM against *Mt*SK [123,128]. Compounds with aromatic modifications at the C5 hydroxyl group of shikimic acid present different inhibition constants against the *Mt*SK and *H. pylori* SK (*Hp*SK). The most potent inhibitor, with an *O*-benzyl moiety, showed *K*_i_ values of 10 µM and 460 nM against *Mt*SK and *Hp*SK, respectively [126]. Rajput and coworkers [129] identified 20 inhibitors, amongst which 5 showed IC_50_ values below 10 µM. The best molecule in the series exhibited an IC_50_ value of 5.1 µM, non-toxic (SI > 10) in vitro cytotoxicity against HepG2 cell line and mitochondrial toxicity; synergistic with rifampicin and bactericidal against *M. tuberculosis* (MIC and Minimum bactericidal concentration of 4 µg/mL each) (Figure 13). Given the promising results for inhibitors of *Mt*SK and the enzyme’s essentiality in the survival of the TB bacillus, more effort should be made on the research of such an attractive target for the development of potential new molecules acting against TB.

### 2.6. 5-Enolpyruvylshikimate-3-Phosphate Synthase (aroA Coding Sequence; EC 2.5.1.19)

The enzyme 5-enolpyruvylshikimate-3-phosphate synthase from *M. tuberculosis* (*Mt*EPSPS), responsible for the penultimate step in the shikimate pathway (Figure 1), is encoded by the *aroA* sequence [130]. The cloning, recombinant expression and (one-step) purification protocols, as well as mass spectrometry analysis and N-terminal sequencing have been reported [130,131]. It has a molecular mass of 47,184 Da and belongs to the enolpyruvyl transferase family that does not require cofactors or metals for catalysis [132].

X-ray diffraction data at 1.81 Å resolution (PDB 2O0E; Figure 14) showed that this monomeric enzyme is an α/β protein consisting of a mixed β-sheet surrounded by α-helices. It is shaped by two domains and with the catalytic/active site near the interdomain crossover segment. Each domain is composed of three copies of a βαβαββ folding unit, with a particular β-sheet in the fourth-strand, which contains parallel and antiparallel strands and all the α-helices are parallel [132,133,134].

Stallings and coworkers [135] deciphered the structure and topological symmetry of the *E. coli* enzyme. They found that each two-domain structure is composed of 6-folding protein units, each one being formed by four-stranded sheets and two parallel α-helices. It was also found that the symmetry and orientation of the two domains make a direct effect on the binding of substrates and inhibitors by a helix macrodipole in *Mt*EPSPS enzyme [133,134].

The catalytic mechanism of *Mt*EPSPS involves the transfer of the enolpyruvyl moiety of PEP to the O5 of S3P [136], through an addition-elimination mechanism. Regarding the addition, it consists on PEP activation by C3 protonation associated with a nucleophilic attack on C2 with a tetrahedral reaction intermediate (THI) that (for a phosphate elimination) leads to a cationic intermediate and finally to 5-enolpyruvylshikimate-3-phosphate (EPSP) (Figure 15) [136,137,138].

Lou and coworkers [138] studied the transition-state for the enzyme-catalyzed hydrolysis reaction measuring kinetic isotope effects. Based on these studies, three possible mechanisms were suggested for EPSP hydrolysis to form S3P and pyruvate catalyzed by *E. coli* EPSPS (assuming that the mechanisms for enzyme-catalyzed hydrolysis are the same as for the acid-catalyzed reaction), in which the reaction could start with: (a) C3 of PEP protonation in a stepwise A_H_*A_N_ mechanism (proton and nucleophile addition occurring in distinct steps with a short-lived cationic intermediate that diffusionally equilibrates with solvent), (b) concerted C3 of PEP protonation and nucleophilic attack on C2 (A_N_A_H_: nucleophile and proton addition occurring simultaneously) or (c) nonactivated nucleophilic attack at C2 in a stepwise A_N_*A_H_ mechanism. Hybrid quantum mechanical/molecular mechanical studies for *E. coli* EPSPS [139] suggest a stepwise mechanism in which protonation of PEP C3 by Glu341 precedes the nucleophilic attack on PEP C2 in the addition mechanism; the breaking of C-O bond of THI to release HPO_4_^2-^ and form the EPSP cation intermediate occurs before proton transfer from PEP C3 to Glu341 (acid/base enzyme catalyst) in the elimination mechanism to form the EPSP product (Figure 16).

EPSPS is the cellular target of *N*-[phosphonomethyl] glycine, the active ingredient of the broad-spectrum, nonselective herbicide glyphosate [140]. The binding of glyphosate to *Mt*EPSPS results in rearrangements of the enzyme’s secondary structure, which are accompanied by a large decrease in solvent access to different regions of the protein [134]. A mechanism for *Mt*EPSPS inhibition by glyphosate was proposed in which inhibitor-induced conformational changes cause a synergistic effect in preventing solvent access to the enzyme’s active site. *Mt*EPSPS exists in an open conformation in the apo form but in a closed conformation when bound to PEP. These conformations were studied by assessing the hydrogen/deuterium exchange properties of free *Mt*EPSPS and *Mt*EPSPS-PEP binary complex using electrospray ionization mass spectrometry (ESI-MS). The enzyme undergoes extensive conformational change upon formation of the PEP binary complex, which seems to favor solvent access at domain 1, while they partially prevent solvent access to domain 2. This may be part of the mechanism of catalysis of the enzyme, which favors both hydration of the substrate-binding site in domain 1 (stabilizing S3P binding) and inducing cleft closure (which controls the entrance of substrate molecules) [132].

Multiple strains resistant to glyphosate were identified not long after the compound’s inhibitory activity was demonstrated in plants [140,141]. Examples in prokaryotes include *E. coli*, in which G96A and A183T mutations in the enzyme rendered the bacteria insensitive to glyphosate [142]. In 2013, Ramachandran and coworkers [143] studied the inhibition of EPSPS in *E. coli* and *M. tuberculosis* with glyphosate, observing that the inhibition values for *Mt*EPSPS were about two orders of magnitude higher (IC_50_ = 260 μM) than for the *E. coli* ortholog (3 μM), suggesting a lower affinity for the mycobacterial enzyme.

The importance of glyphosate in the study of this enzyme has been remarkable; based on the structure of this molecule, hundreds of analogue compounds were developed as candidates to inhibit this protein [139]. Studies proved the capacity of the enzyme to change its natural conformation in the presence of different analogues of glyphosate. Based on that, it was understood that EPSPS could change between open and closed conformations, which seems to be characterized by a decrease in the percentage of β-sheets and increase of the percentage of α-helices [134].

The effects of inorganic phosphate (a product of the reaction) binding to *Mt*EPSPS were assessed by analytical ultracentrifugation, small angle X-ray scattering and circular dichroism techniques, showing that *Mt*EPSPS is in a closed conformation in the presence of inorganic phosphate and in an open conformation in the absence of inorganic phosphate [144]. Replica-exchange metadynamics simulations were employed to assess the effect of structural flexibility on *Mt*EPSPS enzyme function [145]. The authors concluded that these results point to a classical model of conformational selection, in which apo *Mt*EPSPS adopts a conformation that can bind strongly to a ligand and, on the other hand, ligand binding does not induce the enzyme to adopt an appropriate conformation for binding. This structural flexibility is important as a mechanism to adapt to different types of ligands and may be part of the rapid interconversion between different conformations of this protein through the rearrangement of the secondary structure. For example, the enzyme requires the complexation of three water molecules to stabilize the S3P into the binding site [134]. However, the available crystal structure for apo *Mt*EPSPS (PDB 2BJB; manuscript to be published) is likely not the best representative of the conformational states of free enzyme present in solution, because the restricted condition of the crystal generation could produce a flexibility reduction, trapping the protein in a particular state, not in the natural solution state [145].

Numerous compounds that inhibit EPSPS have been proposed for different bacteria, such as *E. coli*, *S. aureus* and *Bacillus cereus* [136,146,147]. Although the sequence similarity between these organisms and *M. tuberculosis* varies between 21-28%, none of these molecules have been tested in *Mt*EPSPS. Kinetic and computational studies are necessary in order to evaluate the susceptibility of *Mt*EPSPS to those compounds.

### 2.7. Chorismate Synthase (aroF Coding Sequence; EC 4.2.3.5)

Chorismate synthase from *M. tuberculosis* (*Mt*CS), the seventh and last enzyme in the shikimate pathway (Figure 1), is encoded by the *aroF* (Rv2540c) gene. This protein has a molecular mass of 41,804 Da [148,149] and belongs to the family of carbon-oxygen lyases. The *Mt*CS structure was determined by X-ray diffraction at 1.72 Å resolution in complex with flavin mononucleotide (FMN) (PDB 2O12) (Figure 17). *Mt*CS is a tetramer composed of two dimers (dimer of dimers) and the monomers belong to the structural α/β class, each monomer containing 13 α-helices and 17 β-sheets and each of them is in contact with the others creating an intricate packing arrangement shaping a core domain composed of a topology of β-α-β sandwich, shown in Figure 17 [150,151].

*Mt*CS catalyzes an unusual 1,4-*anti*-elimination of the 3-phosphate group and the C-(6pro*R*) hydrogen from EPSP forming chorismate and phosphate [150,151]. Circular dichroism spectroscopy, gel filtration and analytical ultracentrifugation results suggest that *Mt*CS exists in a dimer-tetramer equilibrium with a large value for the dissociation constant (50 M), indicating that the enzyme is predominantly in the dimeric form in solution [150].

An interesting feature of CS from different organisms is how reduced FMN (FMN_red_) is obtained, which divides these enzymes into two classes—monofunctional and bifunctional [152]. The CS enzymes from fungi are bifunctional as they display a second enzymatic activity, an NAD(P)H-dependent flavin reductase, which confers them an intrinsic ability to reduce flavin using NAD(P)H. The CS enzymes from plants and *E*. *coli* are monofunctional as they do not possess this activity and are active only in anaerobic conditions in the presence of either chemically or enzymatically reduced flavin. Data on molecular cloning of *aroF*-encoded *Mt*CS, heterologous protein expression and purification, N-terminal amino acid sequencing and mass spectrometry of the recombinant protein indicate the bifunctionality of dimeric *Mt*CS [149].

The enzyme’s flavin reductase activity was characterized, showing the existence of a complex between FMN_ox_ and *Mt*CS. Equilibrium binding of FMN_ox_ pointed to an upper limit of 20 μM for the dissociation constant. NADH binding to *Mt*CS results in a quench in protein fluorescence and a value of 156 μM was obtained for the dissociation constant for binary complex formation. Furthermore, solvent isotope effects and proton inventory results indicated that proton transfer from solvent partially limits the rate of FMN reduction and that a single proton transfer gives rise to the observed solvent isotope effect. Multiple isotope effects suggested a stepwise mechanism for the reduction of FMN_ox_ (distinct steps for hydride and proton transfers) [149].

A new discontinuous method for measuring CS activity has been proposed to be applicable to both monofunctional (anaerobic) and bifunctional (aerobic) enzymes [153]. In short, the method is comprised of in situ production of EPSP by recombinant *Mt*EPSPS, ultrafiltration, addition of FMN, NADH and recombinant *Mt*CS, filtration and detection of chorismate by liquid chromatography coupled with negative electrospray ionization high-resolution tandem mass spectrometry (ESI-HRMS). Although this method allows for determination of activity of a variety of CS enzymes, it would be somewhat difficult to be applicable to the identification of enzyme inhibitors in screening efforts of chemical compound libraries. Structural comparisons of *Mt*CS and other CS enzymes performed by Arcuri and coworkers [151] found that conserved residues (such as His17 and His106) in the active binding site of the bifunctional enzyme are highly conserved amongst the species due to the fact that His106 is near C(2)_O of the Flavin ring and because His17 is near the phosphate group of EPSP, which could probably assist the protonation of the leaving group upon cleavage of C-O bond.

Numerous mechanisms have been proposed to explain the enzymatic reaction that converts EPSP into chorismate [154]. Nevertheless, a three-step mechanism for the CS reaction has been proposed based on quantum mechanics calculations combined with molecular mechanics (QM/MM) [155]. Considering that this reaction has an unusual stereochemistry of the *trans*-1,4-elimination of phosphate, having a proton involved in a non-concerted reaction and a cleavage of an inactivated C-H bond, the mechanism starts with a proton transfer of FMNH_2_ to D339 (R_H3_ transfer), followed by a proton transfer to FMNH (R_H8_ transfer) from EPSP and finally the elimination of phosphate (R_PO_ elimination), as shown in Figure 18 [155,156].

The importance of a deeper understanding of chemical mechanisms is that, based on the structure of some important steps of the reaction, it is possible to developed analogues that can act as competitive or non-competitive inhibitors of the enzyme, blocking catalysis and preventing the formation of the product (chorismate), which is an essential precursor of aromatic amino acids, mycobactins, menaquinones and naphthoquinones. Some strategies based on these studies have proposed analogues, such as (6*S*)-6-fluoro shikimate, (6*R*)-6-fluoro shikimate, (6*S*)-6-fluoro EPSP and 2-(3-(3-((*R*)-3-((*S*)-1-Amino-3-(3-chlorophenyl)-1-oxopropan-2-ylamino)-2-(3-hydroxy-4-methyl-2-nitrobenzamido)-3-oxopropylthio) propylcarbamoyl)phenoxy) acetic acid and other analogues of flavin, such as 2(Z)-2-benzylidene-6,7-dihydroxybenzofuran-3[2H]-ones and the 2′-hydroxy-4′-pentoxy, which inhibited the activity of bifunctional CS enzymes, such as *N. crassa, Plasmodium falciparum, Salmonella typhimurium* and *Streptococcus pneumoniae* [157,158].

To date, no publication has presented a single compound with an inhibitory capacity against *Mt*CS. In 2007, Fernandes and coworkers [159] compared the three-dimensional structure and the geometric docking of the coenzyme FMN and the substrate EPSP amongst *A. aeolicus*, *S. cerevisiae*, *H. pylori* and *S. pneumoniae* (as a model of *M. tuberculosis*) and identified new possible interactions involving Arg111, Gly113 and Ser317 with the EPSP substrate. Since the structure of *Mt*CS was determined a couple of months after this publication, the published analysis did not consider the actual structure of *Mt*CS. For this reason, it is important to rationally develop compounds capable of interfering with Arg111, Gly113 and Ser317 and test them against the enzyme and the organism.

## 3. Future Prospects

The shikimate pathway is essential for the survival of mycobacteria [31]. Given the fact that this metabolic pathway is absent in humans, a better understanding of its component enzymes in *M. tuberculosis* and their chemical mechanisms of action, as well as their three-dimensional structures, should provide a framework on which to base the rational design of selective antimycobacterial agents. Overall, some advances were achieved within the last decade considering the shikimate pathway. Promising inhibitors were developed for some of these enzymes, with attractive MIC values and data on pharmacokinetic properties. This may give one a glimpse on what can be expected as we advance further into drug development. Although not necessarily applicable to the mycobacterial shikimate pathway, a combination with recent and alternative strategies could be explored beyond the ones presented above to target TB.

A multichannel device (iChip), which can simultaneously isolate and grow uncultured soil bacteria, was employed to identify teixobactin, which showed inhibition of *M. tuberculosis* in vitro growth; no drug-resistant strains could be selected [160]. Teixobactin blocks cell wall biosynthesis by binding to peptidoglycan precursors—lipid II (precursor for peptidoglycan biosynthesis) and lipid III (precursor for wall teichoic acid biosynthesis). Its interactions with decaprenyl-coupled lipid intermediates of peptidoglycan and arabinogalactan and its binding to wall teichoic acid precursor contributes to efficient lysis and killing due to digestion of the cell wall by liberated autolysins [160]. It has been pointed out that scale-up synthesis and cost-effective protocols of synthetic teixobactins, a better understanding of its pharmacokinetic properties, extensive in vivo evaluation of efficacy and toxicity and increased barrier to resistance could provide a robust platform for developing a new class of antibiotics for combatting multidrug-resistant bacterial infections [161]. The iChip platform grows and cultures bacteria within a natural environment, in which a soil sample from the environment is sandwiched between two semipermeable membranes and the sample is returned to the environment a short time later. As only approximately 1% of the billions of bacterial species in nature will grow in the lab, the iChip platform can unveil new natural compounds with antibacterial activity.

Cloudbreak is an antibody-drug conjugate platform that combines surface-acting antimicrobial agents with immune engagers in a single molecule that is developed by Cidara Therapeutics (www.cidara.com). The immune system is targeted by stably fusing multiple copies of the antimicrobial agent to the Fc domain of the human IgG1 antibody. The antimicrobial agents are engineered to target conserved regions of the pathogen where mutations often incur major fitness costs. Notwithstanding, the Cloudbreak platform has been used to develop antiviral Fc-conjugates (AVCs) against influenza and, moreover, reports for the development of antibacterials based on this platform are still lacking.

Extracellular *M. tuberculosis* in human TB lesions form biofilms (pellicles) in which the high cell density, cell-cell contacts and different nutrient and oxygen gradients within their interiors result in unique phenotypes, including drug tolerance [162]. A modified version of the culture-based hypoxia model in airtight containers (a phase in which drug-tolerant bacilli evolved) followed by aeration of cell cultures resulted in formation of a pellicle biofilm at the air-liquid interface. Flentie and coworkers [163] identified *R*-8-cyclopropyl-7-(naphthalen-1-ylmethyl)-5-oxo-3,5-dihydro-2*H*-thiazolo [3,2-*a*]pyridine-3-carboxylic acid, which inhibited hypoxia-induced tolerance to isoniazid (INH) in *M. tuberculosis*, prevented selection of drug-resistant strains and reversed INH resistance in *katG* mutants. However, further efforts should be pursued to translate this chemical compound into a chemotherapeutic agent to treat TB infection in human hosts infected with INH-resistant strains of *M. tuberculosis* harboring *katG* mutations [164]. Although the elucidation of the mode of action of this chemical compound may unveil novel targets that can be valuable for the development of new chemotherapeutic agents to treat TB, it has limited therapeutic scope at the present time.

Host-directed therapies have emerged as a novel and promising approach to treat TB, whose modulation of host response is likely to avoid the development of bacterial resistance [165,166]. Unlike conventional antibiotics aimed at targeting the bacteria, host-directed therapy aims at targeting the host immune response either by directly boosting its ability to effectively eliminate the bacterial load or reduce the lung damage associated with granulomatous disease. This strategy has the advantage that molecules would be active against infections taking shelter in the host immune cells and are difficult to target and so might be able to shorten treatment regimens or be effective for individuals with latent TB. Host-directed therapy may be regarded as an adjunct TB therapy to improve the clearance achieved by antibiotics in *M. tuberculosis*.

A new strategy termed PROSPECT (primary screening of strains to prioritize expanded chemistry and targets) was developed in which large chemical libraries (50,000 compounds) are screened against pools of mutant strains depleted in essential bacterial targets (100-150 hypomorphs) and their hypersensitivity are exploited to generate large-scale chemical-genetic interaction profiles (CGIPs). The PROSPECT strategy simultaneously identifies whole-cell active compounds and predicts their mode of action from the primary screening data, which provides putative target information that can help guide hit prioritization [167]. Moreover, a larger target space and new chemical scaffolds may be unveiled by this strategy as compared to whole-cell screening. Using primary CGIPs, it was possible to identify and validate more than 40 new scaffolds against established targets including DNA gyrase, RNA polymerase and the biosynthesis of mycolic acids, folate, tryptophan and the new essential efflux pump EfpA target. Johnson and coworkers [167] have made primary data publicly available (https://broad.io/cgtb) to, in their words, accelerate the discovery of new inhibitor chemotypes and their targets.

In this section we have attempted to give a broad view of expanding experimental strategies to devise new therapies to treat TB. There is also an increased need for expanding the variety and complexity of cell-based assays for biological research and drug discovery. Stem cell-derived cells and tissues have become an increasingly attractive alternative to traditional in vitro and in vivo testing in pharmaceutical drug development and toxicological safety assessment. The Japanese Global Health Innovative Technology Fund proposed a set of criteria for hits and leads identified in screening campaigns of chemical compounds to treat neglected tropical diseases, including TB [168]. These criteria provide clear guidelines for decisions on whether progress from hit to lead and then on into lead optimization. To wrap it up, the concept of focusing on science was exemplified recently when AstraZeneca published a framework founded on project quality and depth of understanding as a key driver of success. This ‘5 Rs’ framework (the right target, the right tissue, the right safety, the right patient and the right commercial potential) can be modified to encompass the influence of mechanistic enzymology in the early stages of drug discovery—right target + right reagents + right assay + right mechanism = right compound [15].

## Figures and Tables

**Figure 1 molecules-25-01259-f001:**
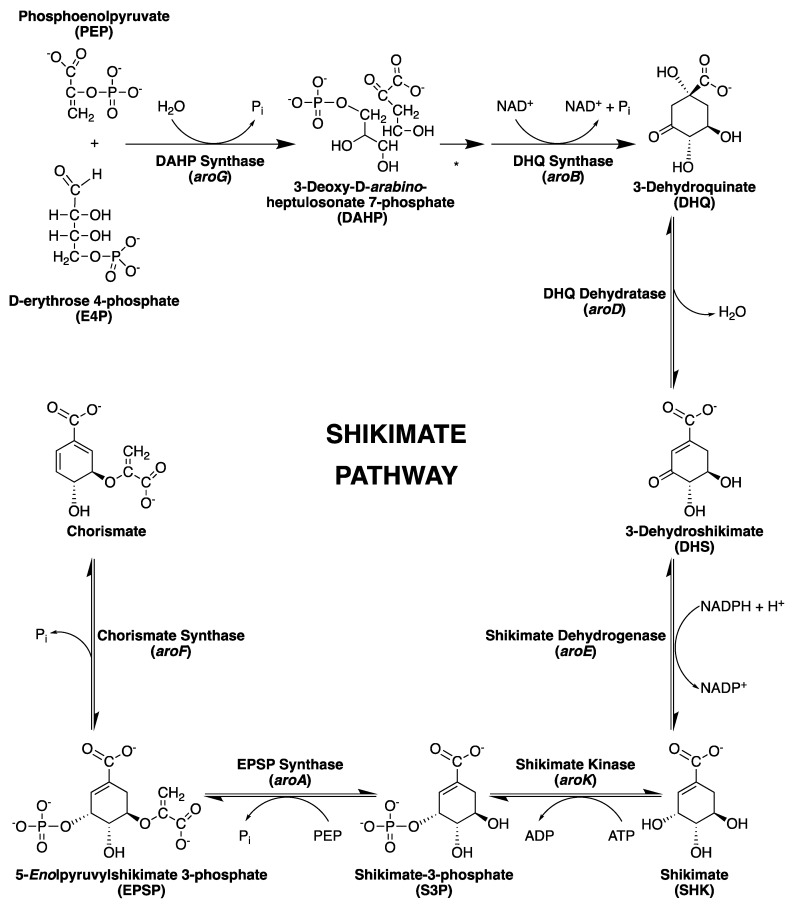
The shikimate pathway. The gene coding for each enzyme is in parenthesis. * Spontaneous cyclization of 3-deoxy-d-arabino-heptulosonate 7-phosphate (DAHP). P_i_ = inorganic phosphate.

**Figure 2 molecules-25-01259-f002:**
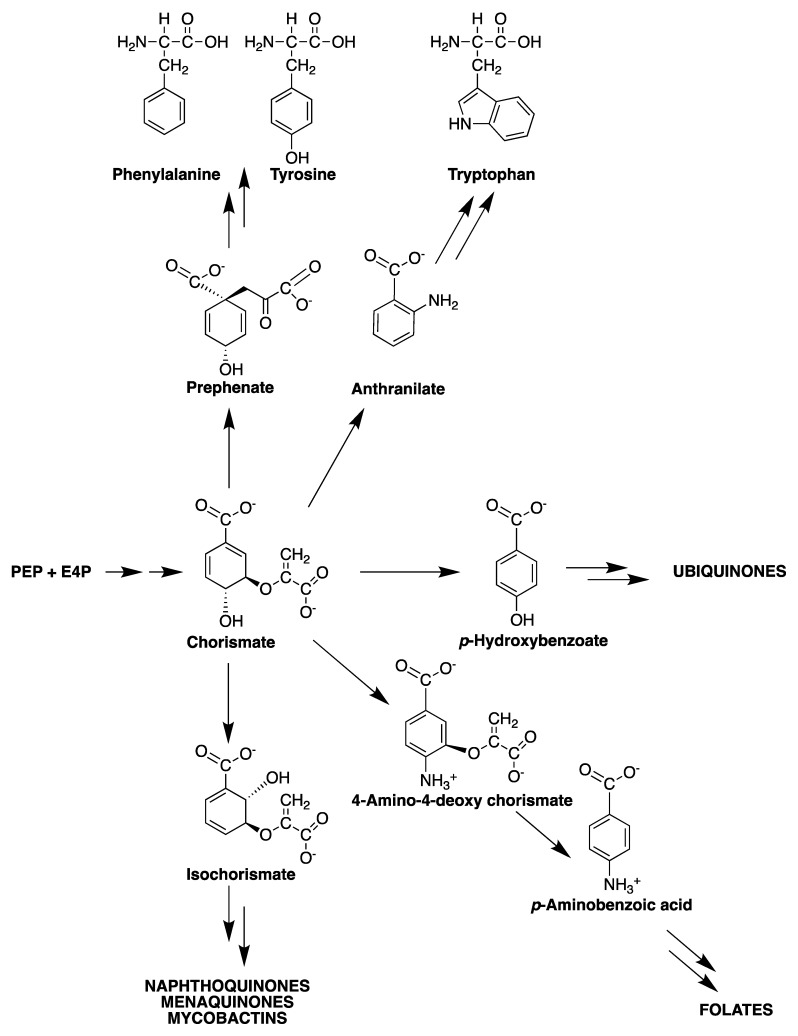
Chorismate: a metabolic node.

**Figure 3 molecules-25-01259-f003:**
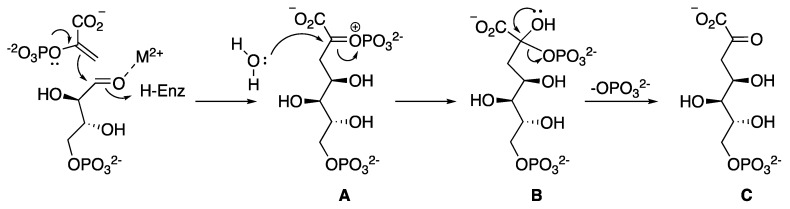
Overall proposed chemical mechanism of condensation of PEP with E4P. H-enz = H-enzyme.

**Figure 4 molecules-25-01259-f004:**
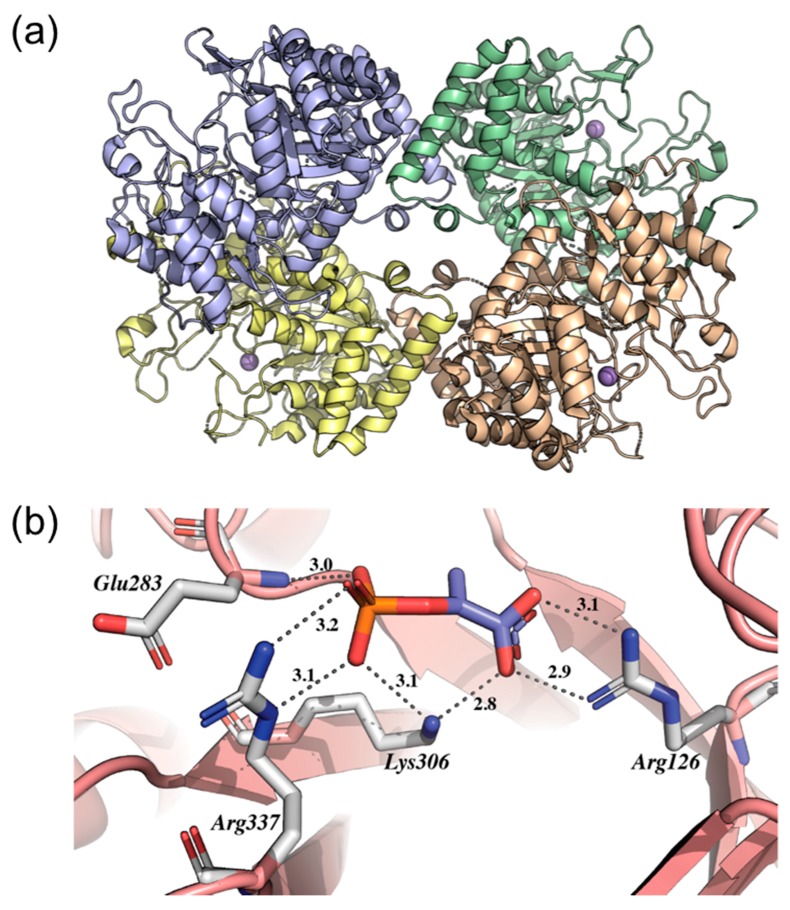
The structure of *Mt*DAHPS. (**a**) Structure of the *Mt*DAPHS tetramer showing monomeric units. Mn^2+^ ions at the *Mt*DAHPS active site are shown as purple spheres. (**b**) Active site of *Mt*DAHPS showing the interactions with the PEP substrate (orange and purple stick model). The images were generated on PyMOL.

**Figure 5 molecules-25-01259-f005:**
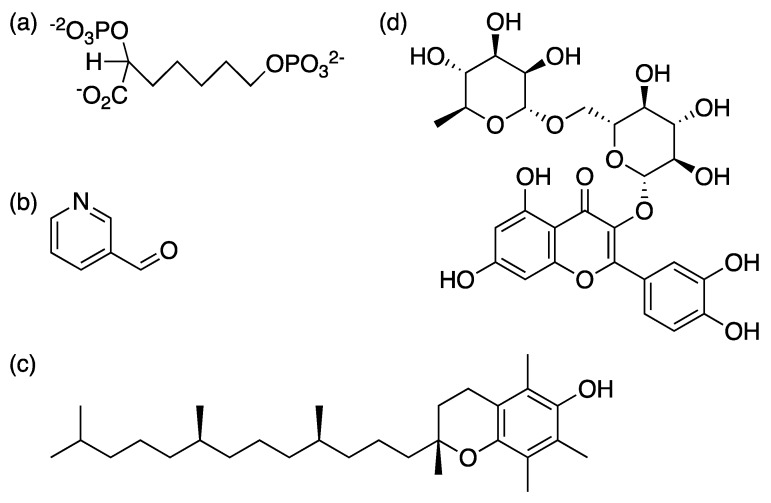
Linear chemical structures of *Mt*DAHPS inhibitors. (**a**) simplified bisphosphate, (**b**) 3-pyridine carboxyaldehyde, (**c**) α-tocopherol, (**d**) rutin.

**Figure 6 molecules-25-01259-f006:**
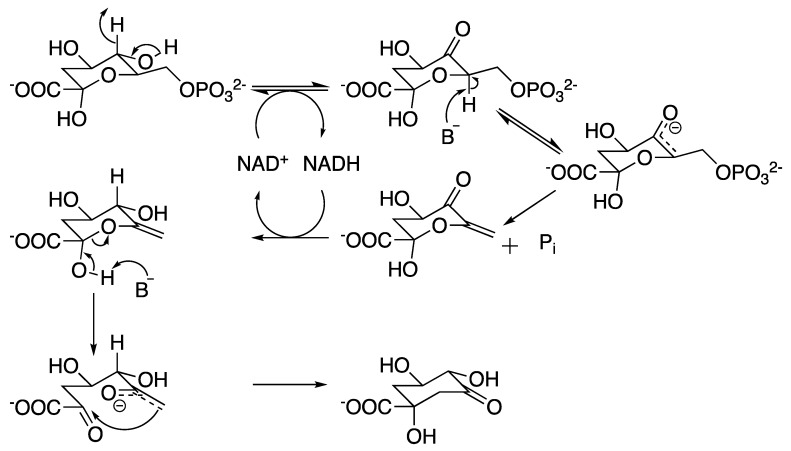
Multi-step reaction catalyzed by *Mt*DHQS.

**Figure 7 molecules-25-01259-f007:**
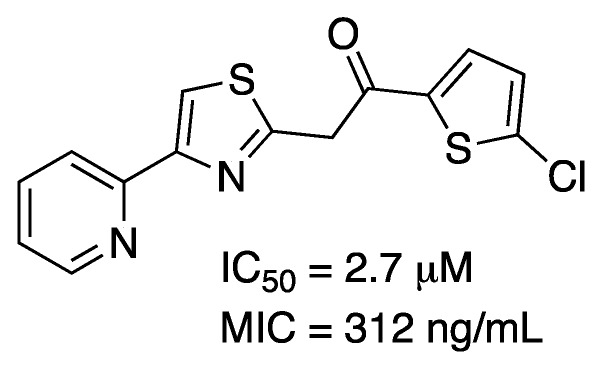
Structure of IMB-T130, a *Mt*DHQS inhibitor.

**Figure 8 molecules-25-01259-f008:**
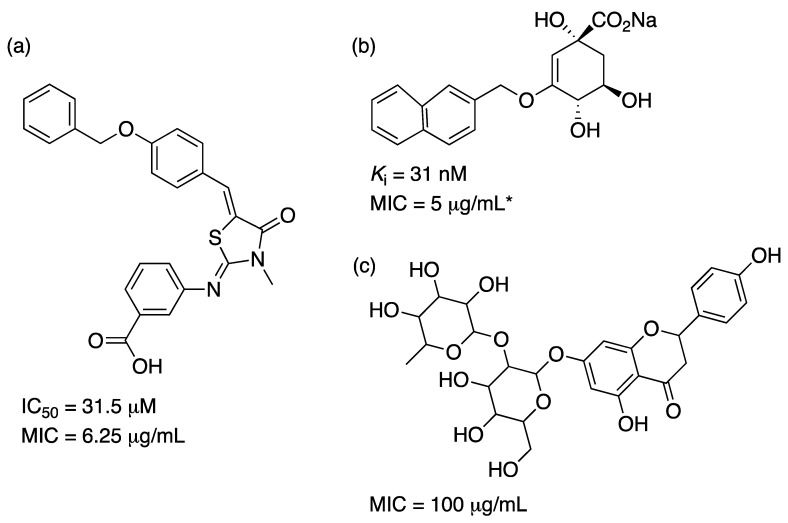
Structure of selected inhibitors of *Mt*DHQase. (**a**) Molecule generated by virtual screening extracted from Petersen and coworkers [69]. (**b**) Example of nanomolar competitive inhibitor extracted from Tizón and coworkers [79]. *MIC obtained with the n-propil ester prodrug. (**c**) Natural occurring molecule (Narigin) found by virtual screening by Lone and coworkers [81].

**Figure 9 molecules-25-01259-f009:**
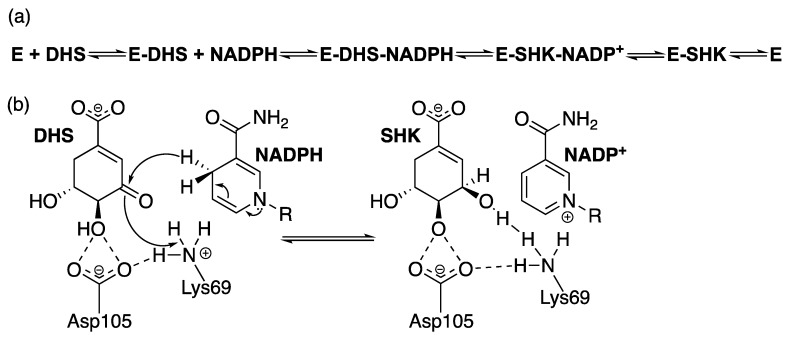
*Mt*SD (**a**) kinetic and (**b**) chemical mechanisms. R represents the ribose, adenosine diphosphate and 2′-phosphate moieties of NADPH.

**Figure 10 molecules-25-01259-f010:**
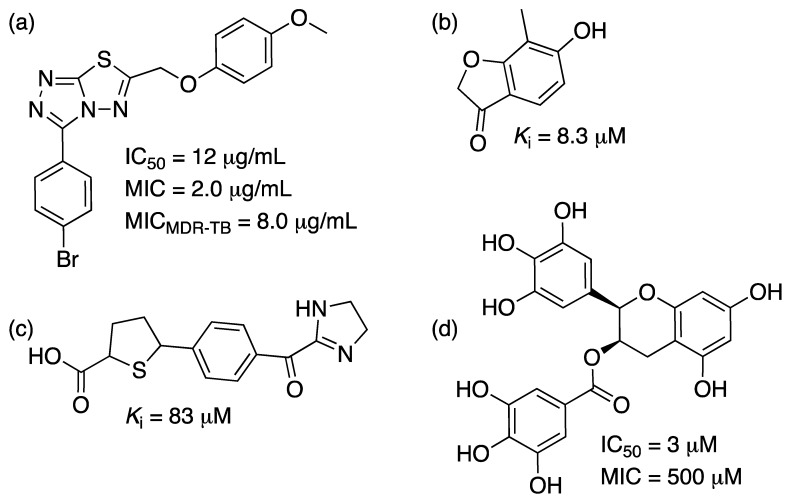
Examples of inhibitors of *Mt*SD. (**a**) Triazolothiadiazole IMB-SD62, inhibitor of *Mt*SD, extracted from Li and coworkers [111,113]. (**b**) and (**c**) *S. aureus* SD inhibitors extracted from Enríquez-Mendiola and coworkers [110]. (**d**) Polyphenolic, inhibitor of *P. putida* SD, extracted from Peek and coworkers [108].

**Figure 11 molecules-25-01259-f011:**
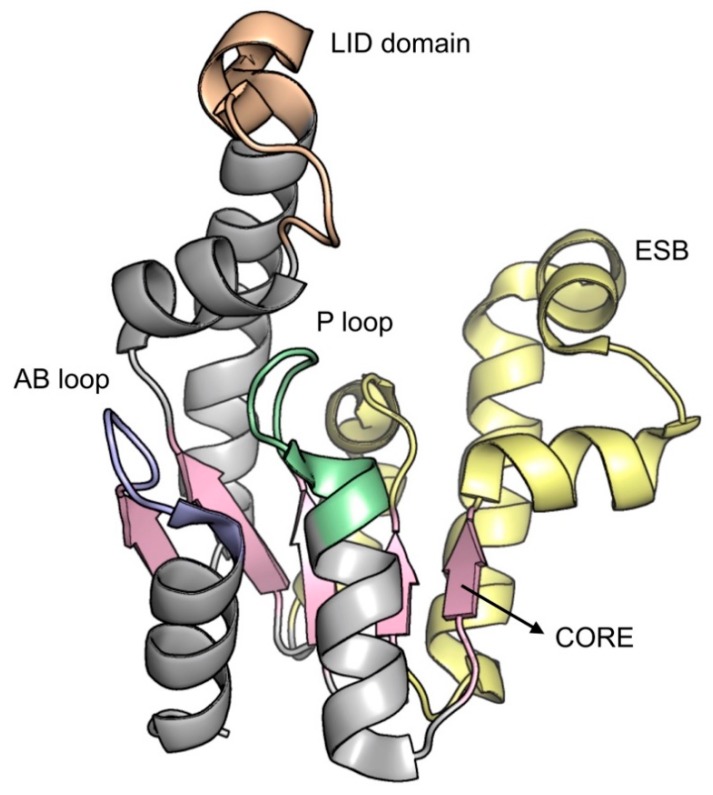
Overall *Mt*SK structure. The image was generated on PyMOL.

**Figure 12 molecules-25-01259-f012:**
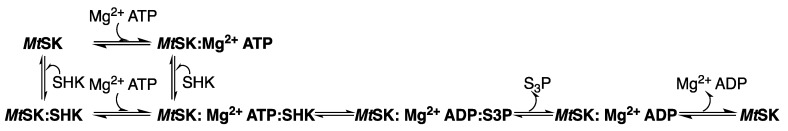
The kinetic mechanism *of Mt*SK.

**Figure 13 molecules-25-01259-f013:**
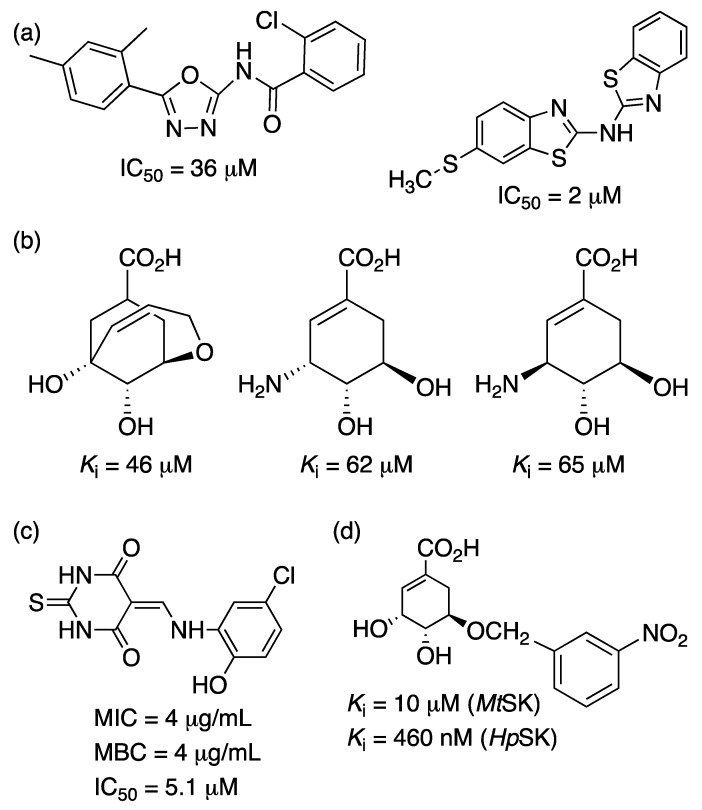
Chemical structures of the top scoring compounds inhibiting *Mt*SK identified from different studies. (**a**) [123,128]; (**b**) [117]; (**c**) [129]; (**d**) [126].

**Figure 14 molecules-25-01259-f014:**
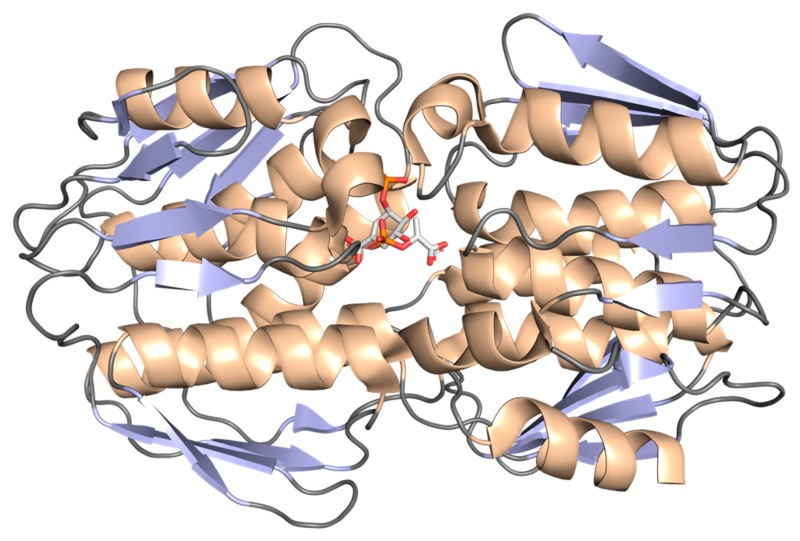
Graphical representation of *Mt*EPSPS tertiary structure in complex with S3P and PEP. The image was generated on PyMOL.

**Figure 15 molecules-25-01259-f015:**
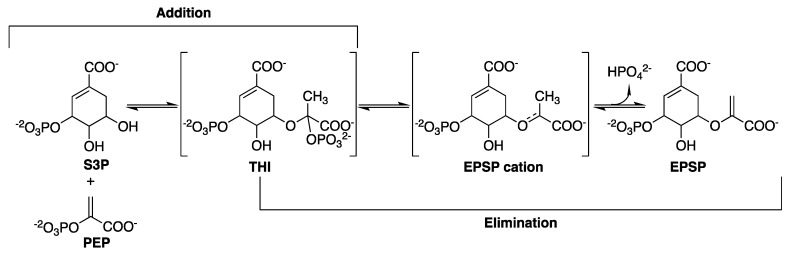
General reaction of EPSPS with the particular tetrahedral intermediate.

**Figure 16 molecules-25-01259-f016:**
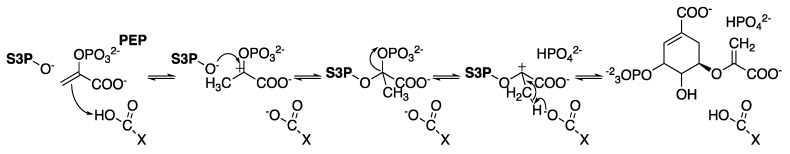
Scheme of the proposed mechanism of catalysis for EPSPS [139].

**Figure 17 molecules-25-01259-f017:**
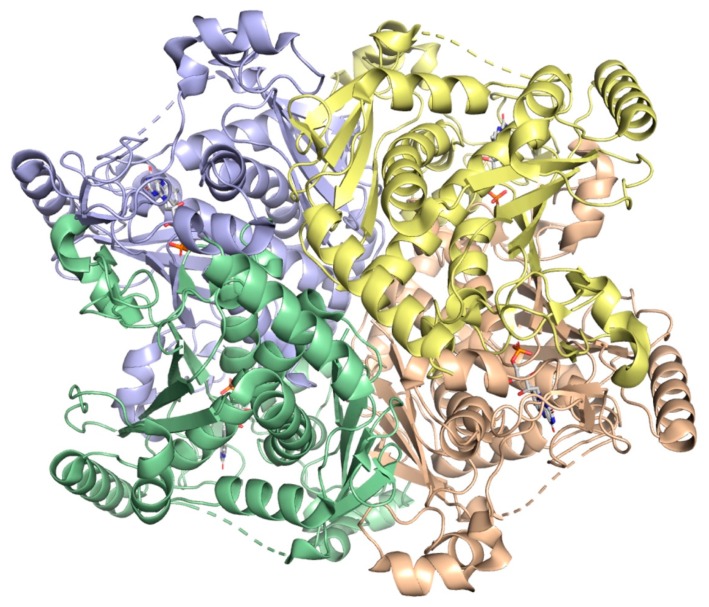
Graphical representation *Mt*CS quaternary structure. The image was generated on PyMOL.

**Figure 18 molecules-25-01259-f018:**
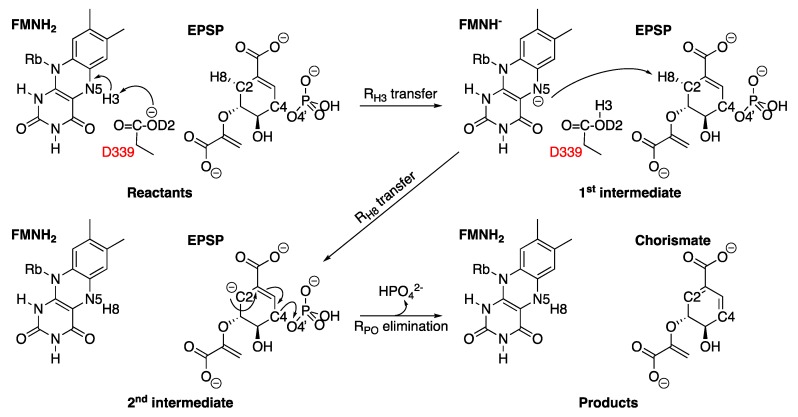
Proposed catalytic mechanism for the CS reaction based on quantum mechanics/molecular mechanics (QM/MM) with the interaction of aspartic acid 339 (D339).

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
