# Peer review of "Mycobacterium tuberculosis Shikimate Pathway Enzymes as Targets for the Rational Design of Anti-Tuberculosis Drugs"

_molecules, 2020, doi:10.3390/molecules25061259_

Round 1

Reviewer 1 Report

The authors described a comprehensive review of the literature and state of teh art in mechanisms and drug discovery for Mtb and the shikimate biosynthesis pathway. This is an important pathway, and plenty of opportunities for further development and advances are presented.

The manuscript is well written and easy to follow with minor changes.

There are comments marked on the attached file, they are mostly related to changes that need to be made in terms of formatting for the chemical structures shown. They need to all use the same formatting, I suggest on ChemDraw “Apply document settings from → ACS Document 1996”, and that this is kept consistent in all figures. There are some issues with some of the arrow pushing, which has been indicated in the attached file.  

Some figures show labeled carbon atoms and this should be corrected - they should be removed (Fig 3 for example). Additionally, several arrows indicating irreversible steps are used when they shouldn't, authors need to check each figure.

Overall, very good review.

Reviewer 2 Report

Please see the docx file.

Reviewer 3 Report

The manuscript entitled “Mycobacterium tuberculosis shikimate pathway enzymes as targets for the rational design of anti-tuberculosis drugs” by Nunes et al. is a comprehensive and interesting paper from a drug design and development perspective for Mycobacterium tuberculosis. The paper has merit for publication, however, it must be reworked substantially before re-considering it for publication.

Here are my comments:

Line 17-27: Please rework the abstract as it is too wordy in its current form.

Section 1.1.: it may contain some information that is redundant and well-known about TB, please try to make it a bit more concise.

Section 1.2. and Section 1.3. is appropriate.

Line 104: Please insert the following reference, relevant to the topic:

https://www.mdpi.com/1420-3049/24/5/892

Section 1.4.: There is a lot of information of enzyme inhibitor kinetics which is general, not specific to the topic. Please limit the discussion of this topic to the absolutely essential elements. As an alternative, merge Sections 1.3 and 1.4 and discuss the topics together.

Section 2. should be shortened substantially (at least by 30%), especially the parts discussing the specific enzymes. The authors should use and incorporate the Figures more in the explanation.

Section 3. is well-written.

References: if possible, the authors should eliminate the older (before 2000) references, as a review should discuss more up-to-date topics.

English use: appropriate.

Round 2

Reviewer 2 Report

In this revised version of the manuscript, the authors tried to address points raised in my previous report and I can state that they succeeded in most of them. Although this is not the best review I have read by far, now it brings at least some useful information toward the medchem design of inhibitors of the enzymes of the Mtb shikimate pathway. Most importantly, the authors added structures of previously reported inhibitors. The question remains how comprehensive the survey for known inhibitors was – the authors in their reply stated that the newly added structures are just ‘Several examples of designed/prepared enzyme inhibitors’.

I was also glad to see that the authors removed or reduced the unnecessary parts of the manuscript which were too general.

On the other hand – this ‘too far away detour’ from the topic remained in the section 3. Future prospects – and my objections toward this passage from my previous report hold. Authors just added the introductory paragraph, where they vaguely postulate that these new technology/gadgets can help in the development of shikimate pathway inhibitors. But from this perspective, why the authors do not describe hundreds of other methods, which can potentially help? Let’s take cryo-EM and its recent advancements as an example. I strongly advice to remove the unnecessary passages of this section – they are off the topic – and have only the final remarks directly related to shikimate pathway and its inhibitors.

When I asked for the methodology which was used to prepare the review, authors replied to me that ‘All the data was obtained from experimental results published in international journals, the great majority of which being indexed to PubMed. General keywords used for the searches were a combination of: “enzyme name”/shikimate pathway, Mycobacterium tuberculosis, enzyme inhibition, (enzyme) drug target, etc. In this review, we have covered publications ranging from 1954 to 2019.’ That’s fine, but I would require this info to be put into the manuscript – it was intended for the readers, not only for one reviewer, right?

To sum up, with minor revisions the manuscript is publishable, although I would expect more from such review.

Reviewer 3 Report

Dear Authors, 

The paper has improved immensely through the revision process and most of my concerns have been addressed by the authors. 

I recommend the paper for publication in Molecules.